# Identifying healthcare experiences associated with perceptions of racial/ethnic discrimination among veterans with pain: A cross-sectional mixed methods survey

**Leslie R. M. Hausmann**[1,2]*, **Audrey L. Jones**[3,4], **Shauna E. McInnes**[1], **Susan L. Zickmund**[3,4]

1 Center for Health Equity Research and Promotion, VA Pittsburgh Healthcare System, Pittsburgh, Pennsylvania, United States of America, 2 Division of General Internal Medicine, Department of Medicine, University of Pittsburgh School of Medicine, Pittsburgh, Pennsylvania, United States of America, 3 Informatics, Decision-Enhancement and Analytic Sciences Center (IDEAS), VA Salt Lake City Health Care System, Salt Lake City, Utah, United States of America, 4 Department of Internal Medicine, University of Utah School of Medicine, Salt Lake City, Utah, United States of America

* leslie.hausmann@va.gov

**Data Availability Statement:** These analyses were performed using DISC study data that are available in identifiable form within the firewall used by the

## Abstract

### Background

Healthcare experiences associated with perceived racial/ethnic discrimination among patients are poorly understood.

### Objective

Identify domains of patient dissatisfaction associated with perceived racial/ethnic discrimination among patients with pain.

### Design

Semi-structured telephone surveys completed in 2013–2015.

### Participants

White, African American, and Latino participants who reported receiving pain management from 25 Veterans Affairs (VA) Medical Centers.

### Main measures

Surveys included open-ended questions about healthcare satisfaction/dissatisfaction and a measure of perceived racial/ethnic-based discrimination while seeking VA healthcare. Binary indicators for ten qualitative domains of dissatisfaction were derived from open-ended questions. We used multilevel models to identify dissatisfaction domains associated with perceived discrimination, adjusting for patient characteristics and site. Within domains associated with discrimination, we identified the most frequent codes and examined whether patients primarily referenced clinical or non-clinical staff in their experiences.

Department of Veterans Affairs (VA). The data are located in a secure research environment known as the VA Informatics and Computing Infrastructure (VINCI). Only aggregate summary statistics and results of our analyses are permitted to be removed from behind the VA firewall in order to comply with general regulatory constraints and VA privacy and data security policies. These restrictions support patient privacy and confidentiality. Those wishing to access the DISC study data are welcome to contact the Associate Chief of Staff for Research and Development at the VA Pittsburgh Healthcare System (Steven Graham, MD: Steven.Graham@va.gov) to discuss the details of the VA data access approval process. A de-identified dataset can be made available upon request pending ethical approval and in accordance with VA guidelines.

**Funding:** This work was supported by the Veterans Integrated Service Network 4 Center for Health Equity Research and Promotion Pilot Research Program (Principal Investigator: LH). The study from which the data were obtained was funded by Department of Veterans Affairs Health Services Research and Development Merit Review (IIR 100144) and Service Directed Research (13-425) awards (Principal Investigator: SZ). Dr. Jones's effort was supported by the National Center for Advancing Translational Sciences of the National Institutes of Health under Award Number UL1TR002538 and KL2TR002539. The views expressed here are those of the authors and do not represent those of the Department of Veterans Affairs or the United States Government. The funders had no role in study design, data collection and analysis, decision to publish, or preparation of the manuscript.

**Competing interests:** The authors have declared that no competing interests exist.

## Key results

Overall, 622 participants (30.4% White, 37.8% African American, 31.8% Latino; 57.4% female; mean age = 53.4) reported a median discrimination score of 1.0 (IQR: 1.0–1.3) on a scale of 1 to 5; 233 (37.5%) perceived any racial/ethnic discrimination in healthcare. Individually, 7 of 10 qualitative domains were significantly associated with perceived discrimination: dissatisfaction with care quality, facilities, continuity of care, interactions with staff, staff demeanor, unresolved pain, and pharmacy services (ps<0.005). In combined models stratified by racial/ethnic group, 3 domains remained statistically significant: poor interactions for Latinos (adjOR = 5.24, 95% CI = 2.28–12.06), negative demeanor for African Americans (adjOR = 2.82, 95% CI = 1.45–5.50), and unresolved pain for Whites (adjOR = 6.23, 95% CI = 2.39–16.28). Clinical staff were referenced more often than non-clinical staff for all domains (interactions: 51% vs. 30%; demeanor: 46% vs. 15%; unresolved pain: 18% vs. 1%, respectively).

## Conclusion

Negative interpersonal experiences and unresolved pain are strong correlates of perceived racial/ethnic discrimination among patients with pain. Future studies should test whether interventions targeting these domains reduce patient perceptions of racial/ethnic discrimination in healthcare.

## Introduction

Feeling as though one has been treated unfairly because of one's membership in a stigmatized group, referred to as perceived discrimination, is associated with less favorable health-related behaviors, mental health, and physical health [1–3]. Perceiving discrimination specifically in healthcare settings has been found to be associated with less positive communication with providers, less satisfaction with care, and greater pain severity, among other unfavorable outcomes [4–9]. Given the sequelae of perceived discrimination, it is important to identify factors that contribute to perceived discrimination in healthcare. Doing so could provide modifiable targets on which to focus interventions to improve experiences of patients from marginalized populations.

Only a handful of studies have investigated the aspects of healthcare experiences that patients are likely to interpret as signs of discrimination [10–12]. In Ross et al.'s qualitative interviews with 12 African American patients, participants noted that discrimination more often comes in the form of "differential treatment" (i.e., being treated differently than other patients) than blatant "mistreatment" (i.e., objectively improper care) [10]. Differential treatment included healthcare staff and clinicians engaging in friendlier, more respectful, and more compassionate verbal and nonverbal communication with people from other races compared to African American patients. Patients also conveyed that healthcare professionals "assumed the worst" about African Americans by applying negative stereotypes (e.g., "drug addicts," faking symptoms). Another study involving 9 African American focus groups found that patients felt discriminated against when providers discredited their symptoms, belittled their health concerns, or did not convey respect [11]. Finally, in qualitative

interviews about satisfaction with healthcare in the Veterans Affairs (VA) Healthcare System, African American Veterans expressed experiencing differential treatment, racial profiling, and being denied treatment [12]. That same study showed qualitative differences in why White and African American Veterans were dissatisfied with pain treatment. Whereas African American Veterans expressed difficulty obtaining pain medications and being treated as drug-seeking, White Veterans were dissatisfied by overly-liberal prescribing of narcotics for pain. However, some White Veterans also felt they had experienced "reverse" discrimination, suggesting that perceived discrimination on the basis of one's race or ethnicity is not restricted to patients of color [12].

Although these prior studies have identified potential aspects of patient experiences that contribute to perceived discrimination, our understanding of factors contributing to perceived discrimination in healthcare settings remains limited. Prior qualitative studies have been small with limited generalizability. Most studies have also focused on the experiences of African American patients, leaving open the question of whether similar factors are associated with perceived discrimination for patients from other racial/ethnic groups.

The objective of this analysis was to identify specific domains of dissatisfaction that are associated with racial/ethnic-based perceived discrimination while seeking VA healthcare among White, African American, and Latino patients with pain. To achieve this objective, we examined qualitative codes, organized into 10 domains of dissatisfaction, and responses to a quantitative measure of discrimination collected as part of the Disparities in Satisfaction with Care (DISC) Study [13]. DISC was a mixed methods survey designed to understand drivers of patient satisfaction/dissatisfaction in a racially and ethnically diverse cohort of male and female patients who received primary care from Veterans Affairs Medical Centers across the United States. We focused the current analysis specifically on the subset of patients who received care for pain management because satisfaction with pain management is an area where racial/ethnic disparities persist and because patients with pain may have the added burden of coping with negative stereotypes (e.g., perceived to be non-compliant, drug-seeking) while presenting for pain treatment. Capitalizing on the richness of the DISC data, we tested for associations between 9 qualitative domains of dissatisfaction and a quantitative measure of perceived racial/ethnic discrimination for all patients receiving care for pain. Given the known racial/ethnic disparities in pain management [14–17], we then looked at associations between the dissatisfaction domains and the quantitative measure of discrimination within White, African American, and Latino groups to gain insights into potential targets for disparity interventions. To further elucidate the observed associations, we examined the most frequently used qualitative codes within each significant domain. When possible, we also described the types of healthcare staff involved in the coded experiences, thus providing further context.

## Methods

### Overview

This analysis examined qualitative domains of dissatisfaction with healthcare and quantitative ratings of perceived discrimination collected as part of a larger study of racial, ethnic, and gender differences in satisfaction/dissatisfaction with care [13]. Participants in the parent study were Veterans sampled, stratified by race and gender, from 25 VA Medical Centers across the United States. Participants completed a semi-structured audio-recorded telephone survey that included open and closed-ended questions about satisfaction/dissatisfaction with healthcare experiences, validated scales assessing constructs that may be associated with patient

healthcare experiences (including perceived discrimination), and clinical and sociodemographic characteristics. As described below, in the current analysis we used qualitative domains of dissatisfaction derived from open-ended questions in the survey as predictors of a quantitative perceived discrimination scale, controlling for select patient characteristics.

Study procedures were approved by institutional review boards at the VA Pittsburgh Healthcare System, VA Salt Lake City, and University of Utah. We followed the Strengthening the Reporting of Observational Studies in Epidemiology guidelines for reporting cross-sectional studies [18]. We also followed the Patient-Centered Outcomes Research Institutes' qualitative and mixed methods standards for reporting the thematic data [19,20].

## Participants

As described elsewhere [13], potential DISC participants included Veterans with an outpatient visit at 25 participating VA medical centers in fiscal years 2012 or 2013 who were identified using administrative records. From those eligible, 90 Veterans from 6 strata (non-Latino White, non-Latino African American, or Latino male; or non-Latino White, non-Latino African American, or Latina female) were randomly selected from each facility [13]. Study sites were geographically diverse and served relatively large proportions of Veterans of color. Potential participants were mailed a study description and were called to confirm eligibility. Following consent, eligible Veterans were surveyed by telephone by a contracted professional survey research organization; surveys were audio-recorded and provided to the research team. Participants received $35 after completing the survey. Data collection took place from June 2013 through January 2015. The overall response rate for the DISC study was 63.3% [13].

The current study focused on White, African American, and Latino DISC participants who met the following additional criteria: 1) responded "yes" to the question, "Have you received pain management from the VA in the last 24 months?"; 2) reported on their satisfaction with pain management; 3) and completed a measure of perceived discrimination (described below). We excluded participants missing data for ≥2 items on the 7-item perceived discrimination measure.

## Semi-structured surveys and thematic analysis

The DISC survey followed a concurrent mixed methods design where Veterans were asked closed-ended Likert scale items and open-ended qualitative questions pertaining to satisfaction/dissatisfaction with specific domains of healthcare experiences, in addition to a series of validated scales and demographic questions [21]. The survey items pertaining to satisfaction/dissatisfaction with healthcare experiences have been previously published [13].

The DISC team coded the open-ended responses from the audio-recorded surveys using the qualitative Editing Approach by Crabtree and Miller [22], which focuses on an open, iterative approach to developing and then applying a codebook. The DISC Principal Investigator [SZ] and the coding team developed the codebook by listening to nearly two hundred surveys as a group and noting codes to capture important content until all sources of satisfaction and dissatisfaction were covered by the qualitative codes. To manage the size of the codebook, qualitative codes were organized into mutually exclusive domains that emerged as dominant categories. Coders met regularly to discuss discrepancies and to refine coding inclusion/exclusion criteria before producing a final master qualitative codebook.

Working initially in teams of two, coders applied the final master codebook to their assigned recordings, each completing the coding independently. The two coders then engaged in an inter-coder reliability adjudication process where they deliberated in order to come to agreement per code. Twenty percent of the interviews were coded using an inter-coder

reliability process. The adjudicated codes were then added into the final qualitative dataset. The full coding team also met throughout the coding process to ensure coding stability and reliability. Given the size of the study, codes and illustrative quotations were entered into a proprietary database powered by Microsoft SQL Server to ensure the effective retrieval of textual data.

## Qualitative domains

Given our focus on perceived discrimination, our analysis concentrated on dissatisfaction codes identified by the DISC coding team (i.e., satisfaction codes were not included). The domains of healthcare experience included: access, quality of care, perception of VA facilities, continuity of care, interactions with clinical and non-clinical staff (e.g., rude, doesn't listen), clinical and non-clinical staff demeanor (e.g., uncaring, stigmatizing), unresolved pain, costs of care, pharmacy services, and non-medical services (e.g., cafeteria, transportation) (Table 1). As in our prior mixed methods research [12,23], we used data transformation to quantify the presence of domains in our sample. Specifically, we assigned binary values to indicate the presence (1) or absence (0) of a domain for each participant. A domain was noted as present if any code within that domain was applied. Given the complexity of modeling the vast number of dissatisfaction codes (n = 213), we used domain indicators (n = 10) as predictors of perceived discrimination in our main analyses. We also examined codes within domains and the type of employee involved in the coded experiences to better understand the nature of the domains that were significantly associated with perceived discrimination (see below).

## Employee type

When participants described dissatisfactory experiences with VA employees, DISC coders noted the type of employee referenced by the participant. We categorized employees as clinical staff (doctors/providers, nurses, nursing staff, surgeons, and pharmacists), non-clinical staff (receptionists/clerks, schedulers, ancillary staff, and volunteers), or unknown/none specified. We assigned binary values to indicate if participants referenced clinical staff (1 = yes, 0 = no) and/or non-clinical staff (1 = yes, 0 = no) for each dissatisfaction code (e.g., rude, unconcerned) within the domains.

## Primary outcome

The primary outcome in this analysis was perceived racial/ethnic discrimination while seeking VA healthcare, which was assessed in the DISC Study using an adaptation of the Everyday Discrimination Scale [5,24,25]. Specifically, participants were asked how often they experienced 7 types of unfair treatment while seeking healthcare because of one's race or color (e.g., "When getting healthcare, how often were you treated with less respect than other people because of your race or color?" 1 = never, 2 = rarely, 3 = sometimes, 4 = most of the time, 5 = always). For participants answering at least 6 of the 7 items, we calculated an overall discrimination score as the average of non-missing items; participants missing 2 or more items were excluded. Because the distribution was positively skewed, with most participants reporting "never" for all items, we categorized participants as having any racial/ethnic-based perceived discrimination (i.e., those who reported experiencing 1 or more item at least rarely) versus none.

## Covariates

We included as study covariates several sociodemographic and clinical characteristics assessed on the DISC survey that could be associated with patient experiences with care and/or

**Table 1. Domains of dissatisfaction with healthcare experiences and related codes derived from semi-structured surveys about patient satisfaction.**

| Domains and codes* | Sample Quotes |
|---|---|
| **Access** | |
| Scheduling | "It's kind of difficult to get appointments." (Scheduling) |
| Timeliness | |
| High patient volume / crowded | "I couldn't get in to see my doctor. I had to go to a local doctor." (Felt forced to go to non-VA facility) |
| Red tape / bureaucracy | |
| Unable to email/call provider | |
| Lack of providers | |
| Lack of treatment options | |
| Felt forced to go to non-VA facility | |
| General mentions of poor care | |
| VA not prepared to address issues specific to women | |
| Felt forced to go to emergency room | |
| **Quality of care** | |
| Dissatisfied with treatment plan | "I am with her for 5 minutes and she shoves me out the door." (Didn't take time) |
| Unsatisfactory diagnosis | |
| General mentions of poor access Didn't take time | |
| Unresolved medications | "He kept insisting that he couldn't give me any more medicine, and I kept insisting that I hurt." (Did not receive medication) |
| Incompetence / unknowledgeable | |
| Did not receive medication/medical equipment | |
| Not thorough | "Getting the doctors and technicians to listen to me as a patient and allowing me to contribute to my healthcare is difficult." (Involvement in decisions) |
| Negligent | |
| Involvement in decisions | |
| Didn't protect privacy | |
| Tries to avoid VA services | |
| Inaccurate diagnosis | |
| **Facilities** | |
| Parking | "I have to go 45 minutes to an hour earlier than my appointment just so I can get parking." (Parking) |
| Location | |
| Difficult to navigate | |
| Size | "Right now it's under a bunch of construction and it's hard to find your way around." (Renovations) |
| Equipment | |
| Old building | |
| Renovations | |
| Unclean/unsanitary | |
| Inaccessible for persons with disability | |
| **Continuity of care** | |
| Staff turnover | "She's never followed up with me, or even called me back either." (Follow-up) |
| Follow-up | |
| VA to VA coordination | "There's been numerous occasions when my primary care has put me through for referral . . . and I never got a phone call with an appointment." (Inadequate referrals) |
| Inadequate referrals | |
| Not seen by regular provider | |
| Doesn't review history | |
| General mentions of poor continuity | |

(*Continued*)

**Table 1.** (Continued)

| Domains and codes* | Sample Quotes |
|---|---|
| **Interactions with staff** | |
| Rude/condescending/hostile | "They don't listen to me about what helps and what doesn't help." (Doesn't |
| Doesn't listen | listen) |
| Non-informative | |
| Poor relationship | |
| General mentions of poor interactions | |
| **Staff demeanor** | |
| Unconcerned/uncaring | "She was mean, mean. She accused me of selling my medications." (Stigma) |
| Stigma | |
| Dishonest/untrustworthy | "When I go in there, it's like I'm just a number." (Treats Veteran like a |
| Not accommodating or helpful | number) |
| Treats Veteran like a number | |
| Unprofessional | |
| Inattentive | |
| **Unresolved pain** | |
| Perceived drug-seeking | "I'm getting something for pain, but it's not treating the problem." |
| General mentions of unresolved pain | (General) |
| **Costs** | |
| Service connectedness | "[I cannot receive] dental and vision. I am not qualified for it." (Service |
| Copays | connectedness) |
| Having to pay | |
| **Pharmacy services** | |
| Pharmacy ordering | "If you go to the pharmacy, you have to pull a number and the waiting time is no sooner than 30 minutes and you can wait an hour before you see them. Then you have to wait again." (Pharmacy ordering) |
| **Non-medical services** | |
| Transportation | "I'm somebody who can't walk right now. It's so far apart that I need to call for transport and that takes a while." (Transportation) |

*Dissatisfaction domains that emerged from audio-recorded surveys are shown in bold, with individual codes listed blow each domain. Codes present for at least 5% of all participants shown. General codes are statements of dissatisfaction that mention the domain without providing additional detail (e.g., Dissatisfaction with interactions—General: "It's just that communication is bad.")

perceived discrimination. Specifically, we included self-reported age, gender, race/ethnicity (in analyses not stratified by race/ethnicity), education, perceived health status (fair/poor versus good/very good/excellent), and self-reported depression. To account for exposure to the VA healthcare system, we also included the number of VA outpatient visits in the year prior to the survey, which was drawn from VA administrative records.

## Statistical analyses

We used Stata version 14 to conduct all statistical analyses [26]. We first compared the sociodemographic characteristics for participants with any versus no perceived discrimination, using chi-square tests and t-tests for all categorical and continuous variables, respectively. For subsequent analyses, we imputed missing sociodemographic data for 9 (1.4%) participants using a multiple imputation package in Stata (mi estimate) that averaged the model estimates across 5 imputed datasets and produced pooled standard errors according to Rubin's rules [27].

Next, we individually tested the association of each qualitative domain with any (vs. no) perceived discrimination using mixed effect logistic regression models. Each model included main effects for race/ethnicity, gender, sociodemographic covariates (age, education, health status, depression, and number of VA outpatient visits in the prior year), and a random effect for study site. We set the criteria for statistical significance at $p<0.005$ after applying a Bonferroni correction to account for multiple hypothesis testing (i.e., testing the associations of 10 domains of dissatisfaction with perceived discrimination).

Next, we tested a final adjusted multivariable model containing all qualitative domains that were significantly associated with any (vs. no) perceived discrimination when tested separately. We tested the final model in the full sample and separately for each racial/ethnic group. In sensitivity analyses, we reran the analyses with mean discrimination scores using linear mixed models.

To further understand the qualitative domains that were statistically significantly associated with perceived discrimination in multivariable models, we examined the frequency of individual codes within each significant domain among participants who reported any perceived discrimination. For the most frequently mentioned codes within each domain, we identified illustrative quotes and calculated the percentage of participants who referred to clinical and non-clinical employees when describing those experiences.

## Results

### Sample characteristics

Of the 1,222 DISC participants, 716 (58.6%) received pain treatment in the past 24 months and 634 (51.9%) reported on experiences with VA pain management. After excluding participants with ≥2 missing items on the measure of perceived discrimination (n = 6, 0.9%) and participants who reported a race/ethnicity other than non-Latino White, non-Latino African American, or Latino (n = 6, 0.9%), the analytic sample included 622 Veterans (Fig 1). Reflecting the stratified design of the DISC study [13], just over half of those included in this analysis were women (57.4%) and about one-third were African American (37.8%) or Latino (31.8%). The sample had substantial physical and mental health service needs, with 42.1% rating their health status as fair or poor, 52.0% reporting a history of depression, and, on average, having 24.0 (standard deviation = 25.3) outpatient visits in the prior 12 months.

In the 622 patients included in the analysis, the median rating of perceived discrimination was 1.0 (IQR: 1.0–1.3) on a 1 to 5 scale; 233 patients (37.5%) reported any (vs. no) perceived racial/ethnic discrimination while seeking VA healthcare. Patients who perceived any discrimination were significantly different from those who had perceived no discrimination in several ways (Table 2). Specifically, patients who perceived any (vs. no) discrimination were more likely to be African American (48.1% vs. 31.6%), rate their health status as poor or very poor (49.4% vs. 37.8%), have a history of depression (63.5% vs. 45.2%), and have more outpatient visits in the prior year (mean = 28.5 vs. 21.2).

### Qualitative domains associated with racial/ethnic-based perceived discrimination

The 10 qualitative domains varied widely in prevalence, with 567 participants (91.2%) expressing dissatisfaction with access to care and only 54 participants (8.7%) expressing dissatisfaction with non-medical services (Table 3). Patients who perceived any (vs. no) discrimination were more likely to mention dissatisfaction with 7 of the 10 healthcare domains, including quality of care (83.3% vs. 64.3%), facilities (76.8% vs. 65.0%), continuity of care (67.4% vs. 49.6%),

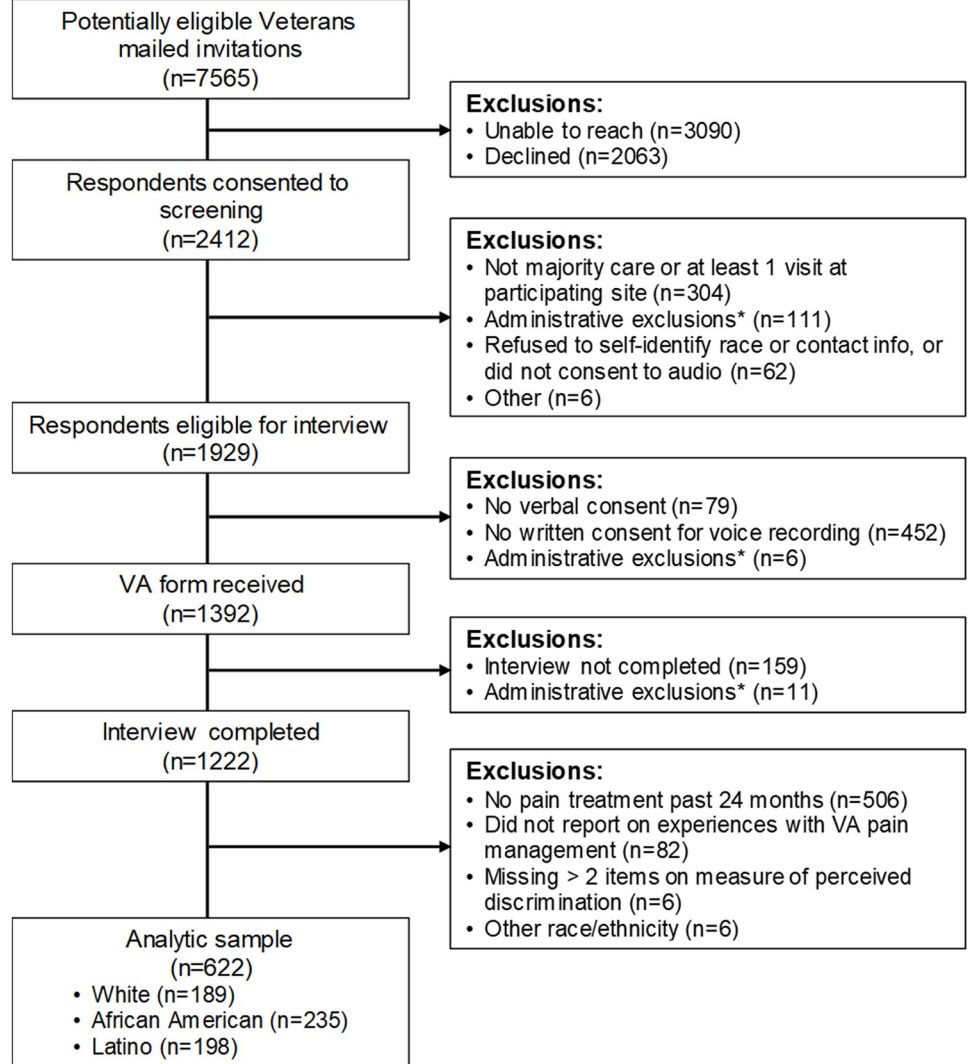

**Fig 1. Participant recruitment and study enrollment.** Exclusion criteria were applied sequentially. *The predominant reason for administrative exclusion was that a recruitment cell was filled by the time a potential respondent worked their way through the multi-step recruitment process.

interactions with staff (76.4% vs. 41.9%), staff demeanor (64.8% vs. 35.5%), unresolved pain (45.5% vs. 24.2%), and pharmacy services (29.6% vs. 19.5%).

When the above 7 domains were included as predictors in a single multivariable model adjusted for covariates, 2 domains—interactions with staff and staff demeanor—were the only domains of dissatisfaction that remained statistically significant at p<0.005 (interactions: adjusted odds ratio [adjOR] = 2.86, 95% confidence interval [CI] = 1.81–4.50; demeanor: adjOR = 2.30, 95% CI = 1.50–3.54; Table 4).

Models stratified by racial/ethnic group indicated that dissatisfaction with interactions with staff was significant at p<0.005 only among Latino patients (adjOR = 5.24, 95% CI = 2.28–12.06), whereas dissatisfaction with staff demeanor was only significant among African American participants (adjOR = 2.82, 95% CI = 1.45–5.50; Table 5). In addition, unresolved pain was statistically associated with perceived discrimination among White participants (adjOR = 6.23, 95% CI = 2.39–16.28).

**Table 2. Characteristics of patients who received healthcare for pain management, stratified by whether they reported any (vs. no) racial/ethnic-based perceived discrimination while receiving healthcare.**

| Characteristics | Total (n = 622) | | No PD (n = 389) | | Any PD (n = 233) | | p-value* |
|---|---|---|---|---|---|---|---|
| | N | % | N | % | N | % | |
| Race/ethnicity | | | | | | | <0.001 |
| Non-Latino White | 189 | 30.4 | 138 | 35.5 | 51 | 21.9 | |
| Non-Latino African American | 235 | 37.8 | 123 | 31.6 | 112 | 48.1 | |
| Latino | 198 | 31.8 | 128 | 32.9 | 70 | 30.0 | |
| Female gender | 357 | 57.4 | 220 | 56.6 | 137 | 58.8 | 0.584 |
| Age (years) | | | | | | | 0.032 |
| 18–39 | 115 | 18.5 | 76 | 19.5 | 39 | 16.8 | |
| 40–59 | 290 | 46.7 | 166 | 42.7 | 124 | 53.5 | |
| 60+ | 216 | 34.8 | 147 | 37.8 | 69 | 29.7 | |
| Education level | | | | | | | 0.772 |
| ≤High school/GED | 73 | 11.8 | 48 | 12.4 | 25 | 10.8 | |
| Trade school/some college | 300 | 48.5 | 184 | 47.7 | 116 | 50.0 | |
| ≥College graduate | 245 | 39.6 | 154 | 39.9 | 91 | 39.2 | |
| Fair/poor health status | 262 | 42.1 | 147 | 37.8 | 115 | 49.4 | 0.005 |
| Depression | 322 | 52.0 | 176 | 45.2 | 146 | 63.5 | <0.001 |
| Number of VA outpatient visits in the past 12 months (mean, sd)† | 24.0 | 25.3 | 21.2 | 22.9 | 28.5 | 28.3 | <0.001 |

*p-value obtained from chi-square test

†p-value obtained from t-test

Abbreviations: GED, general education diploma; PD, perceived discrimination; sd, standard deviation; VA, Veterans Affairs

**Table 3. Frequency and percentage of domains of dissatisfaction with healthcare experiences, stratified by whether they reported any (vs. no) racial/ethnic-based perceived discrimination while seeking healthcare.**

| Domains of Healthcare Dissatisfaction | Total (n = 622) | | No PD (n = 389) | | Any PD (n = 233) | | p-value* |
|---|---|---|---|---|---|---|---|
| | N | % | N | % | N | % | |
| Access | 567 | 91.2 | 352 | 90.5 | 215 | 92.3 | 0.458 |
| Quality of care | 444 | 71.4 | 250 | 64.3 | 194 | 83.3 | <0.001 |
| Facilities | 432 | 69.5 | 253 | 65.0 | 179 | 76.8 | <0.001 |
| Continuity of care | 350 | 56.3 | 193 | 49.6 | 157 | 67.4 | <0.001 |
| Interactions with staff | 341 | 54.8 | 163 | 41.9 | 178 | 76.4 | <0.001 |
| Staff demeanor | 289 | 46.5 | 138 | 35.5 | 151 | 64.8 | <0.001 |
| Unresolved pain | 200 | 32.2 | 94 | 24.2 | 106 | 45.5 | <0.001 |
| Costs | 153 | 24.6 | 88 | 22.6 | 65 | 27.9 | 0.136 |
| Pharmacy services | 145 | 23.3 | 76 | 19.5 | 69 | 29.6 | 0.001 |
| Non-medical services | 54 | 8.7 | 26 | 6.7 | 28 | 12.0 | 0.071 |

*p-value obtained from mixed effect logistic regression of perceived racial/ethnic discrimination while seeking healthcare. Each model included fixed effects for the qualitative domain and patient sociodemographic characteristics (race/ethnicity, gender, age, education level, health status, depression, and number of outpatient visits at Veterans Affairs Medical Centers in the past 12 months); and a random effect for study site. Domains that were significant after applying a Bonferonni correction for multiple comparisons (p<0.005) were included in a final combined model.

**Table 4. Final multivariable model containing domains of dissatisfaction associated with any (vs. no) racial/ethnic-based perceived discrimination while seeking healthcare (n = 622).**

| Predictors | Odds Ratio | Confidence Interval | P-value* |
|---|---|---|---|
| Quality of care | 1.12 | 0.67–1.87 | 0.669 |
| Facilities | 1.49 | 0.97–2.30 | 0.069 |
| Continuity of care | 1.13 | 0.73–1.75 | 0.594 |
| Interactions with staff | 2.86 | 1.81–4.50 | <0.001 |
| Staff demeanor | 2.30 | 1.50–3.54 | <0.001 |
| Unresolved pain | 1.69 | 1.12–2.54 | 0.013 |
| Pharmacy services | 1.53 | 0.99–2.36 | 0.056 |

*P<0.005 is considered statistically significant after applying Bonferroni correction for multiple comparisons. Estimates were obtained using mixed effect logistic regression with any (vs. no) racial/ethnic-based perceived discrimination while seeking health care as the outcome. The model included fixed effects for the qualitative domains and patient sociodemographic characteristics (race/ethnicity, gender, age, education level, health status, depression, and number of outpatient visits at Veterans Affairs Medical Centers in the past 12 months), and a random effect for study site.

## Sensitivity analyses

We observed similar patterns of results for the overall sample in linear models that treated perceived discrimination as a continuous outcome. Patients who expressed dissatisfaction with interactions with staff and staff demeanor had perceived discrimination scores that were 0.15 and 0.20 points higher, respectively, when compared to patients who did not express dissatisfaction in these domains (interactions with staff: b = 0.15, 95% CI = 0.05–0.25; demeanor: b = 0.20, 95% CI = 0.10–0.29; S1 Table).

## Thematic analysis of codes of dissatisfaction with staff interactions, staff demeanor, and unresolved pain

We further examined prominent codes for the three domains that were significantly associated with any perceived discrimination for any racial/ethnic group: interactions with staff, staff demeanor, and unresolved pain. In the interactions domain, the most frequent code was encountering healthcare employees who were rude, condescending, or hostile. One Latino man noted: "The second they give you the script on why you can't get an appointment, they

**Table 5. Final multivariable models containing dissatisfaction domains associated with any (vs. no) racial/ethnic-based perceived discrimination while seeking healthcare, stratified by racial/ethnic group.**

| Predictors | Whites (n = 189) | | African Americans (n = 235) | | Latinos (n = 198) | |
|---|---|---|---|---|---|---|
| | Odds Ratio | CI | Odds Ratio | CI | Odds Ratio | CI |
| Quality of care | 0.48 | 0.13, 1.73 | 1.51 | 0.71, 3.21 | 1.22 | 0.48, 3.12 |
| Facilities | 1.36 | 0.45, 4.12 | 1.61 | 0.85, 3.05 | 1.85 | 0.83, 4.14 |
| Continuity of care | 1.60 | 0.55, 4.69 | 1.26 | 0.63, 2.52 | 1.02 | 0.48, 2.19 |
| Interactions with staff | 2.05 | 0.66, 6.38 | 2.05 | 1.02, 4.11 | 5.24* | 2.28, 12.06 |
| Staff demeanor | 2.86 | 0.99, 8.26 | 2.82* | 1.45, 5.50 | 1.34 | 0.61, 2.92 |
| Unresolved pain | 6.23* | 2.39, 16.28 | 2.02 | 1.05, 3.89 | 0.46 | 0.21, 1.02 |
| Pharmacy services | 2.56 | 1.08, 6.08 | 1.01 | 0.47, 2.16 | 1.47 | 0.65, 3.32 |

*Statistically significant after applying Bonferroni correction for multiple comparisons (p-value<0.005). Estimates were obtained using mixed effect logistic regression with any (vs. no) racial/ethnic-based perceived discrimination while seeking health care as the outcome. Models included fixed effects for the qualitative domains and patient sociodemographic characteristics (gender, age, education level, health status, depression, and number of outpatient visits at Veterans Affairs Medical Centers in the past 12 months), and a random effect for study site. Analyses were conducted separately for White, African American, and Latino participants.

get very confrontational when you ask questions and they think you're being difficult." Another frequent code was feeling like employees were not listening to them. A White woman described the experience: ". . .getting the doctors and technicians to listen to me as a patient and allowing me to contribute to my healthcare is difficult. I'm very aware of the injury I sustained and the problems I have related to it and there have been a lot of times that doctors don't care to listen to me, basically." The third most frequent code in this domain was feeling as though employees were uninformative. An African American woman explained: "I was satisfied, but they never told me why they quit giving me the injections, they never explained to me why they stopped. . . One day I went to the clinic and there was a new fellow there, and he said that he couldn't give me the injections and that they couldn't give them to me, and it was over- that was gist of the explanation to me." Among the 233 Veterans who perceived any discrimination, these codes were expressed by 56.2%, 31.3%, and 19.7% respectively.

For staff demeanor, the most frequent code was that employees were unconcerned or uncaring. One African American man stated flatly: "Forgive me for even saying this, but you are treated like waste. The level of treatment that you receive from those people that you have to sit back and deal with that are not sensitive to your needs because they have no idea what is going on with you." Another frequent code was feeling stigmatized. One Latina woman recounted a conversation with a provider: "I had another doctor that told me, my pain management doctor, told me that the reason why I was in so much pain is because I was here in the U.S. and because this is not my birth place. I needed to just move back home." The third most frequent code in the domain of staff demeanor was distrust. One African American man shared: "I have had my last two treatments at private physicians because the doctors that were [at the VA] are gone and I do not trust people there anymore." These codes were expressed by 29.6%, 23.6%, and 15.0%, respectively, by Veterans with perceived discrimination.

For unresolved pain, most statements were grouped under a general code. For instance, one White woman recalled: "There have been numerous times that I have went to my primary care physician because the pain was obviously getting worse and. . . she always wanted to talk around the issue instead of directly about the issue and trying to improve the issue." A Latina woman shared: "They gave me this pain pill and it's not working and that's all they can give me. I'm not asking for stronger but can you do something else? It's interfering with my job and my lifestyle." One sub-theme, perceived drug-seeking, was common among White participants with perceived discrimination. One white man pleaded: "You can't neglect people because they need it and can't treat them like they were drug addicts." General themes of unresolved pain and perceived drug-seeking were expressed by 39.9% and 12.5% respectively.

The patterns of codes were generally similar across racial/ethnic groups with some subtle nuances (Table 6). For example, many of the codes were more frequent among White participants than among African American or Latino participants, especially rudeness, not listening, treating Veterans like a number, general statements about unresolved pain, and being perceived as drug-seeking. Also, the distribution of individual codes pertaining to staff demeanor were different among African American and Latino participants. For example, stigma, dishonest/untrustworthy, and unhelpful codes came up more frequently for African American participants, whereas the relatively infrequent codes of unprofessional and uninviting/unwelcoming demeanor came up more often for Latino participants (Table 6).

## Types of employees referenced when describing dissatisfaction with staff interactions, staff demeanor, and unresolved pain

Among participants who reported any perceived discrimination, 66%, 56%, and 18% mentioned a specific type of employee when expressing dissatisfaction in the domains of staff

**Table 6. Frequency of dissatisfaction codes in the domains of interactions with staff, staff demeanor, and unresolved pain expressed by patients who perceived any discrimination, stratified by racial/ethnic group.**

| | All groups (n = 233) | | Whites (n = 51) | | African Americans (n = 112) | | Latinos (n = 70) | |
|---|---|---|---|---|---|---|---|---|
| **Codes** | N | % | N | % | N | % | N | % |
| **Interactions with staff** | | | | | | | | |
| Rude/condescending/hostile | 131 | 56.2 | 32 | 62.8 | 54 | 48.2 | 45 | 64.3 |
| Doesn't listen | 73 | 31.3 | 23 | 45.1 | 29 | 25.9 | 21 | 30.0 |
| Uninformative | 46 | 19.7 | 10 | 19.6 | 23 | 20.5 | 13 | 18.6 |
| Poor relationship | 33 | 14.2 | 7 | 13.7 | 17 | 15.2 | 9 | 12.9 |
| General mentions of poor interactions | 29 | 12.5 | 8 | 15.7 | 15 | 13.4 | 6 | 8.6 |
| **Staff demeanor** | | | | | | | | |
| Unconcerned/uncaring | 69 | 29.6 | 16 | 31.4 | 31 | 27.7 | 22 | 31.4 |
| Stigma | 55 | 23.6 | 16 | 31.4 | 27 | 24.1 | 12 | 17.1 |
| Dishonest/untrustworthy | 35 | 15.0 | 9 | 17.7 | 18 | 16.1 | 8 | 11.4 |
| Unhelpful | 29 | 12.5 | 9 | 17.7 | 16 | 14.3 | 4 | 5.7 |
| Treats Veteran like number | 28 | 12.0 | 11 | 21.6 | 10 | 8.9 | 7 | 10.0 |
| Inattentive | 21 | 9.0 | 5 | 9.8 | 10 | 8.9 | 6 | 8.6 |
| Unprofessional | 19 | 8.2 | 3 | 5.9 | 8 | 7.1 | 8 | 11.4 |
| Uninviting/unwelcoming | 16 | 6.9 | 3 | 5.9 | 6 | 5.4 | 7 | 10.0 |
| **Unresolved pain** | | | | | | | | |
| General | 93 | 39.9 | 27 | 52.9 | 43 | 38.4 | 23 | 32.9 |
| Perceived drug-seeking | 29 | 12.5 | 12 | 23.5 | 12 | 10.7 | 5 | 7.1 |

*Dissatisfaction domains that emerged from audio-recorded surveys are shown in bold, with individual codes listed blow each domain. Codes present for at least 5% of participants with perceived discrimination (n = 233) are shown. General codes are statements of dissatisfaction that mention the domain without providing additional detail (e.g., Dissatisfaction with interactions—General: "It's just that communication is bad.")

interactions, staff demeanor and unresolved pain, respectively. Except for the code of rudeness, patients referred to clinical staff more often than non-clinical staff across the most frequent codes in these domains (interactions: 51% vs. 30%, respectively; demeanor: 46% vs. 15%; unresolved pain: 18% vs. 1%; see Fig 2).

## Discussion

This mixed methods analysis used data from the DISC Study to identify aspects of patient experiences associated with perceived racial/ethnic discrimination in healthcare among a large, diverse sample of patients being treated for pain. It is important to understand experiences of perceived discrimination in healthcare settings for patients with pain, specifically, as experiences of discrimination have been linked to increased pain sensitivity, pain severity, disability, and development of chronic pain [8,9,28–31]. In addition, there are well-documented racial/ethnic disparities in both the clinical impact and treatment of pain, for which discrimination may be a contributing factor [14,17,32]. While we considered a wide range of qualitative domains as potential correlates of perceived discrimination in healthcare, only those that pertained to interpersonal interactions and unresolved pain emerged as statistically significant in fully adjusted models stratified by racial/ethnic group. Other aspects of care, including the condition of healthcare facilities, care continuity, and costs, were not associated with perceived discrimination for White, African American, or Latino patients.

We observed variation in the types of domains that were associated with perceived discrimination across racial/ethnic groups, with dissatisfaction in interactions with staff being

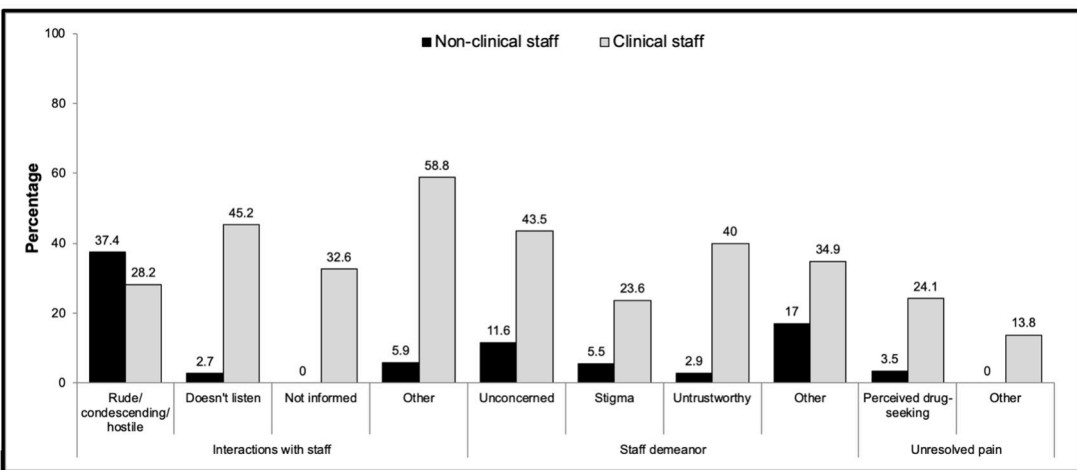

**Fig 2. References to clinical and non-clinical staff when describing dissatisfaction with staff interactions, staff demeanor, and unresolved pain.**

associated with perceived discrimination among Latinos and dissatisfaction with staff demeanor being associated with perceived discrimination among African Americans. The overarching pattern of negative experiences in interpersonal domains for African American and Latino racial/ethnic groups support conclusions from prior qualitative studies that perceptions of discrimination in healthcare among patients from minority populations manifest, in large part, from poor interpersonal interactions with clinical and non-clinical staff [10–12]. Our results are also consistent with research on microaggressions, which are commonplace indignities or insults that are directed, often unintentionally, at members of marginalized groups [33,34]. The types of interpersonal interactions associated with perceived discrimination among African American and Latino patients in the current study suggest that patients were experiencing microaggressions from healthcare employees [35]. It is noteworthy that, even though negative interactions with staff and encountering staff with negative demeanor were equally or more prevalent among White patients, such interpersonal slights were only associated with perceived racial/ethnic discrimination among minority patients.

Another nuance observed in this study was that unresolved pain was associated with perceived discrimination only among White patients. One interpretation of this novel finding is that stigma and frustration surrounding pain management may be a salient issue for White patients, whereas negative interpersonal interactions with healthcare employees in general may be more salient experiences among African American and Latino patients. Given the exploratory nature of this mixed methods analysis, additional research is needed to better understand the nuances in themes we observed as correlates of perceived discrimination across racial/ethnic groups.

While an earlier paper published from the DISC cohort has reported on quantitative ratings of healthcare satisfaction [13], the current mixed methods analysis draws on rich qualitative codes to help identify factors associated with perceived discrimination in healthcare for patients receiving pain management. At first glance, the higher prevalence of perceived discrimination among African American Veterans is at odds with previously published DISC findings of few racial/ethnic differences in quantitative ratings of healthcare satisfaction. As in prior work, we find that Likert satisfaction ratings alone do not fully capture negative healthcare experiences such as those perceived as discriminatory [12,36]. The qualitative data analyzed in the current analysis provide a richer lens to understand the more nuanced aspects of

healthcare encounters that contribute to negative experiences, including those attributed to one's race/ethnicity.

While the study design does not permit causal inference, our thematic analysis of the domains that were associated with perceived discrimination in this study (i.e., staff interactions, staff demeanor, unresolved pain) highlights areas deserving of future research and potential intervention. First, Latino patients who perceived discrimination were likely to have interactions with healthcare employees who were rude, did not listen to patients, or did not provide the information patients desired. African American patients who perceived discrimination were likely to describe healthcare employees who appeared unconcerned about the patient's wellbeing, made the patient feel stigmatized, or were untrustworthy. Additionally, White patients who perceived discrimination felt they were being accused of illicit behavior and substance abuse. Future research is needed to determine if targeting these aspects of interpersonal interactions improve patient experiences or reduce perceptions of discrimination in healthcare settings. Such strategies could include expanded provider training in pain management and addiction stigma, as well as active listening and other communication skills among the healthcare delivery workforce. Other strategies could involve institutional changes to provide more time in the clinical encounter for pain management, and to support respectful and empowered communication between healthcare employees and patients.

Second, it is troubling that, among our sample of patients with pain, over one in five who perceived discrimination described healthcare experiences in which they felt overtly stigmatized. The prevalence of perceived discrimination observed in this study is within the wide range observed in prior studies that have assessed perceived racial/ethnic discrimination in VA healthcare settings [4,5,37–39]. Although the overall prevalence varies across studies depending on the targeted patient population and the measure used to assess perceived discrimination [5], a common pattern observed across the literature is that racial/ethnic discrimination is more frequently reported by racial/ethnic minority patients compared to White patients, as was the case for our study. Amidst a growing number of studies documenting the presence and impact of unconscious biases in the healthcare setting, our findings indicate that instances of blatant biases still occur [40]. It is not possible from the current study to determine if stigma related to addiction, race/ethnicity, or other factors are direct causes of perceived discrimination. And yet, the high prevalence of stigma themes among persons receiving treatment for pain underscores the magnitude of this problem in healthcare.

Additional work is needed to identify effective strategies to reduce stigma for patients with pain. Some promising avenues for future research might include 1) initiatives that promote principles of diversity and inclusion across all aspects of healthcare delivery [41], 2) education for providers and staff in unconscious biases [42] and misperceptions about biological differences between racial/ethnic groups among healthcare providers [32], and 3) deconstructing institutionalized processes that contribute to differential power, privilege, and outcomes across groups [43]. It is also important for future research in this area to determine if interventions aimed at healthcare discrimination could ameliorate persisting racial/ethnic disparities in pain management and outcomes.

Third, our analysis suggests that, contrary to prior qualitative studies that suggested that most perceptions of discrimination emanated from interactions with non-clinical staff [10,12], dissatisfying interpersonal interactions in our sample more often occurred with clinical staff. One potential reason for the different findings is that our study focused exclusively on patients treated for pain. We posit that disputes with providers over pain management create tension in the clinical encounter where provider biases could emerge or patients could attribute unsatisfying treatment plans to their race or ethnicity [28,44,45]. The one exception to our observation of negative interactions with healthcare providers related to rude, condescending, or

hostile interactions, which were slightly more often associated with non-clinical than with clinical employees. These findings suggest that addressing behaviors that are associated with perceived discrimination in healthcare settings need to include healthcare employees in both clinical and non-clinical roles.

We note the following study limitations. First, our sample includes Veterans of the United States military who received care for pain from VA medical facilities, thereby limiting generalizability to other patient samples and healthcare systems. However, the domains of dissatisfaction with healthcare experiences that emerged were not unique to pain management or the VA healthcare system. A second limitation is that, because the larger study was about patient satisfaction in general and targeted primary care patients, the survey did not include clinical measures of pain. A third limitation is the cross-sectional nature of the study, which precludes making causal inferences about associations between dissatisfaction with certain domains and perceived discrimination. A related limitation is that we did not ask patients specifically about their experiences with discrimination and what was associated with them, so the correlation between qualitative domains and perceived discrimination could be due to an unobserved variable or reverse causality. Finally, the retrospective nature of the surveys, which covered patient experiences with all VA healthcare in the past year, makes the data subject to recall biases. Prospective studies in which surveys are linked to a single healthcare encounter and explicitly focus on experiences perceived as discriminatory may provide more precise estimates of associations of healthcare domains and perceived discrimination. We also note study strengths. Drawing from semi-structured surveys with over 600 patients from three racial/ethnic groups drawn from 25 VA medical centers nationwide, our analysis is one of the largest and most comprehensive investigations, to date, of healthcare experiences that are associated with perceptions of racial/ethnic discrimination among patients.

## Conclusions

Despite the well-established negative association between racial/ethnic-based perceived discrimination and health outcomes, perceived discrimination is not routinely identified or addressed in the context of healthcare. As shown in this large, mixed methods survey of Veterans seeking pain management from the VA, interpersonal aspects of patient healthcare experiences are strong correlates of racial/ethnic-based perceived discrimination in the healthcare setting across multiple racial/ethnic groups. Future studies should investigate the variation in themes associated with perceived discrimination across White, African American, and Latino patients and test whether interventions targeting these domains reduce patient perceptions of racial/ethnic discrimination in healthcare.

## Supporting information

**S1 Table. Multivariable linear regression model containing domains of dissatisfaction associated with racial/ethnic-based perceived discrimination while seeking healthcare (n = 622).** *P<0.005 is considered statistically significant after applying Bonferroni correction for multiple comparisons. Estimates were obtained using mixed effect linear regression with rating of racial/ethnic-based perceived discrimination while seeking health care (range: 1–5) as the outcome. The model included fixed effects for the qualitative domains and patient sociodemographic characteristics (race/ethnicity, gender, age, education level, health status, depression, and number of outpatient visits at Veterans Affairs Medical Centers in the past 12 months), and a random effect for study site.
(TIFF)

## Acknowledgments

The authors thank Kelly Burkitt, PhD, VA Pittsburgh Healthcare System, for serving as site Principal Investigator for the Disparities In Satisfaction with Care (DISC) Study, Nicole Beyer, MA, and Nichole K. Bayliss, PhD, for serving as Project Coordinators, and the team of research assistants and qualitative coders who carried out the recruitment and qualitative coding for the data used in this paper.

## Author Contributions

**Conceptualization:** Leslie R. M. Hausmann, Audrey L. Jones, Shauna E. McInnes, Susan L. Zickmund.

**Data curation:** Audrey L. Jones, Susan L. Zickmund.

**Formal analysis:** Audrey L. Jones, Susan L. Zickmund.

**Funding acquisition:** Leslie R. M. Hausmann, Susan L. Zickmund.

**Investigation:** Leslie R. M. Hausmann, Audrey L. Jones, Shauna E. McInnes, Susan L. Zickmund.

**Methodology:** Leslie R. M. Hausmann, Audrey L. Jones, Shauna E. McInnes, Susan L. Zickmund.

**Project administration:** Shauna E. McInnes.

**Resources:** Susan L. Zickmund.

**Supervision:** Leslie R. M. Hausmann, Susan L. Zickmund.

**Validation:** Susan L. Zickmund.

**Writing – original draft:** Leslie R. M. Hausmann, Audrey L. Jones.

**Writing – review & editing:** Leslie R. M. Hausmann, Audrey L. Jones, Shauna E. McInnes, Susan L. Zickmund.

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
