## [Decision Letter · Decision Letter 0]

16 Apr 2020

PONE-D-20-03894

Identifying Healthcare Experiences Associated with Perceptions of Racial/Ethnic Discrimination among Patients with Pain: A Mixed Methods Study

PLOS ONE

Dear Dr. Hausmann,

Thank you for submitting your manuscript to PLOS ONE. After careful consideration, we feel that it has merit but does not fully meet PLOS ONE’s publication criteria as it currently stands. Therefore, we invite you to submit a revised version of the manuscript that addresses the points raised during the review process.

We would appreciate receiving your revised manuscript by May 31 2020 11:59PM. To enhance the reproducibility of your results, we recommend that if applicable you deposit your laboratory protocols in protocols.io, where a protocol can be assigned its own identifier (DOI) such that it can be cited independently in the future. For instructions see: http://journals.plos.org/plosone/s/submission-guidelines#loc-laboratory-protocols

We look forward to receiving your revised manuscript.

Kind regards,

M Barton Laws

Academic Editor

PLOS ONE

Additional Editor Comments (if provided):

Please pay particular attention to Reviewer 1's comments about clear explanation of the methodology, and clarity and appropriateness of inference.

2. Please include additional information regarding the survey or questionnaire used in the study and ensure that you have provided sufficient details that others could replicate the analyses. For instance, if you developed a questionnaire as part of this study and it is not under a copyright more restrictive than CC-BY, please include a copy, in both the original language and English, as Supporting Information. Moreover, please include more details on how the questionnaire was pre-tested, and whether it was validated.

Reviewers' comments:

Reviewer's Responses to Questions

**Comments to the Author**

1. Is the manuscript technically sound, and do the data support the conclusions?

Reviewer #1: Partly

Reviewer #2: Yes

2. Has the statistical analysis been performed appropriately and rigorously? 

Reviewer #1: Yes

Reviewer #2: Yes

3. Have the authors made all data underlying the findings in their manuscript fully available?

Reviewer #1: No

Reviewer #2: Yes

4. Is the manuscript presented in an intelligible fashion and written in standard English?

Reviewer #1: Yes

Reviewer #2: Yes

5. Review Comments to the Author

Reviewer #1: This manuscript presents findings from a national, cross-sectional survey of VA patients stratified by race/ethnicity (non-Hispanic white vs Hispanic vs black). Authors code open-ended telephone survey responses and then conduct regression analyses to identify correlates of perceived discrimination overall and by veteran race/ethnicity. The focus on veterans with pain is a strength, because racial/ethnic disparities in pain management have been well documented and pain is a common topic of patient dissatisfaction. As detailed below, there are several areas that dampen my enthusiasm: the study design is not clear early on to readers, a clearer focus on implications for pain management (including possibly analyzing pain as a covariate) and more tempered conclusions appropriate to the limitations of cross-sectional data. If these concerns can be addressed, the manuscript has potential to make a meaningful contribution to the literature and advance knowledge.

Specific comments:

1. Title. Overall study design and procedures are not clear based on the abstract and intro; I re-read abstract several times and it was not clear until well into methods section that this study is a survey with both closed-ended items and (semi-structured) open-ended interview questions. Emphasizing the “survey” and de-emphasizing the “mixed methods” aspect throughout manuscript will make study design clearer for readers. Specifically, specifying the population and study design in title will decrease potential for reader confusion. Title should be, “Identifying healthcare experiences … among veterans with pain: a cross-sectional mixed methods survey.”

2. Abstract. Edit to make study design clearer for readers. In DESIGN, consider replacing the term “telephone interviews” with “telephone survey.” In MAIN MEASURES, make it clearer that perceived discrimination (dependent variable) and open-ended questions (which were coded and transformed to derive independent variables) were both collected during the same survey interview. I appreciate the mixed methods aspect but the quantitative component is predominant in this study and should drive how study is described.

3. Abstract. Final sentence of CONCLUSION needs to be more qualified / cautious. It is too strong to conclude definitively from this study that findings “underscore the need for interventions…” In addition, implications may be specific to veterans with pain.

4. Introduction, next-to-last paragraph, page 6. Intro discusses the need to understand correlates of perceived discrimination to improve care generally. However, given the known racial/ethnic disparities in pain management, study findings specifically have potential to generate hypotheses/insights that can help to address racial/ethnic disparities in pain management. Making this point explicitly in the intro (and briefly citing literature on racial/ethnic disparities in pain mgmt) would strengthen study justification and make study more compelling for readers.

5. Introduction, final paragraph. Again, study design is difficult to discern from this paragraph. Describing the survey design before the qualitative coding and using the term “mixed methods survey” instead of “mixed methods study” will help clarify design for readers.

6. Participants, page 9. Adding more detail about patient pain would be helpful given the study focus on patients w pain. It is also plausible that pain & pain management would differ for veterans with vs without perceived discrimination. Did parent study collect data on pain (eg pain numeric rating scale, PEG, pain type/location; study mentions that DISC collected administrative VHA data)? If so, pain-specific variables should be included as covariates or at least summarized in Table 2. If parent study did not collect data on patient pain, this should be acknowledged as a limitation. Did the QOL measures include pain-specific items that could be reported separately?

7. Page 10. Specify whether the coders working in teams of two independently applied the master codebook to each interview or whether they coded together as a team.

8. Table 1. This is a nice table. Since “unresolved pain” is a specific code within the quality of care domain, it would be helpful for results to specify proportion of patients for whom this code was present. This might be reported in table 3 or in the manuscript where table 3 is discussed. would also be helpful to note in parentheses after each example the code(s) that example represents within each domain.

9. Table 1. All the examples under “staff demeanor” relate to pain. Since demeanor was one of the significant correlates of perceived discrimination, it would be helpful to describe in results section the prevalence of pain-specific codes for this domain (either quantitatively, qualitatively, or both). A parallel analysis for the other significant domain (interaction w staff) would also be helpful. This additional detail would help convey to readers how salient pain-related comments are among the survey responses, and by extension, the extent to which pain-specific concerns may correlate w perceived discrimination.

10. Pages 20-21. Very interesting that sub-analyses found that “interaction w staff” was correlated w dissatisfaction only for Hispanic vets, while “demeanor” was correlated w dissatisfaction only for black vets. However, on page 20-21 all the examples or “interaction w staff” came from black vets while almost all the examples of “demeanor” came from Hispanic vets. Why is this?

11. Related to point 10 above, I more detailed results about the differences in findings by veteran race/ethnicity would strengthen the study. As mentioned in the intro, discrimination experienced by Hispanic vets is understudied relative to black vets, so including large proportion of Hispanic vets is a particular strength of this study. Although Tables S1 and S3 present some differences by veteran race/ethnicity and there are obvious differences by race/ethnicity, these differences are mostly glossed over in the main manuscript. For Table S3, may be helpful to see differences across all the themes (not just top 3). More nuanced discussion of qualitative and quantitative differences by race/ethnicity (including differences that may not have meet the very strict corrected p value threshold) would help generate hypotheses about how perceived discrimination differs for black vs Hispanic vs NH white veterans, which in turn would increase study impact.

12. Finally, related to point 11 above, manuscript needs to report more detail about perceived discrimination by NH white veterans and the factors/domains correlated with perceived discrimination in these patients compared to Hispanic and black veterans. Introduction (page 6) makes a point to cite prior studies about experience of “reverse discrimination” and states that discrimination can be perceived by vets of any race/ethnicity. However, results and discussion sections only focus on Hispanic and black veterans.

13. Discussion, page 22. Can authors put the prevalence of any perceived discrimination in their sample (37.5%) in context? Is this higher, lower, or similar to rates reported in other studies of veterans or non-veterans? A quick pubmed search found some articles that may be relevant (e.g. Sorkin, D. H., et al. "Racial/ethnic discrimination in health care: Impact on perceived quality of care." Journal of General Internal Medicine 25(5): 390-396).

14. Second sentence of discussion, line 423. “discrimination” should be “perceived discrimination.”

15. Third sentence of discussion, line 428. Both references (ref 8, 24) are specific to patients with sickle cell disease. References more relevant to this study population would be helpful.

16. Discussion, page 22-23. The finding that interpersonal interactions are consistent drivers of perceived discrimination across race/ethnicity is very interesting and important. I appreciate that the “overarching pattern of negative experiences in interpersonal domains … [was similar for] all racial/ethnic groups.” However, more detail about the specific differences and variation across the 3 racial/ethnic groups studied should be included in discussion, even if similarities outweigh differences. As noted above, the intro highlighted the inclusion of Hispanic vets and potential ‘reverse discrimination’ among NH white vets as strengths of this study. Thus, readers will be expecting results and discussions related to these two topics.

17. Discussion, page 23 line 452. Please replace the term “soft skills” with communication skills, remove the scare quotes, and remove communication skills from the list at the start of that sentence. All the examples provided in this paragraph are specific types of communication skills. It is inappropriate to use terminology that implies communication skills are less important or more subjective than other clinical skills.

18. Discussion. On page 25, the authors appropriately discuss limitations of this cross-sectional survey, particularly inability to assess causation or control for unmeasured confounding. However, the paragraph starting on line 443 (pages 23-24) seems to ignore these limitations by suggesting that study results directly support specific future interventions which all implicitly assume that experiencing the themes identified causes perceived discrimination. This paragraph should be much more cautious about results interpretation and next steps and include discussion of both potential ‘causal’ and ‘non-causal’ explanations/interpretations of study findings. Based on data presented, reverse causation is also plausible (i.e. patients who have experienced past discrimination are more likely to report/remember/perceive discrimination in the VA).

19. Page 23, the sentence starting on line 454 (“Disruptive innovations in healthcare models… should also be explored.”) should be deleted. Even setting aside the inability of this study to evaluate causal associations, the data presented does not support or suggest that these types of interventions are likely to reduce perceived discrimination. If anything, the data suggest such “disruptive” system-level innovations would not reduce perceived discrimination, because systems domains (facilities, nonmedical aspects of care, continuity of care, access) were not associated w perceived discrimination in the final model.

20. Similarly, the two sentences starting on page 465 (“not only should such initiatives…in this study”) should be deleted because they do not follow from the data presented and implicitly interpret correlations as causal associations. The study did not address unconscious bias or beliefs about biological difference between races, or how presence of these staff attitudes might relate to the qualitative themes.

21. As noted in prior comments, it would strengthen discussion to talk about whether findings and further efforts to reduce perceived discrimination among veterans has the potential to ameliorate documented racial/ethnic disparities in pain management.

22. Authors have already published data from parent study on positive patient comments (JGIM 2018;33:305-331), where they found no consistent differences in correlates of satisfaction by veteran race/ethnicity. Would be helpful to briefly mention the results of that analysis to put the current findings related to negative comments in context. If correlates of veteran satisfaction are similar across racial/ethnic groups but correlates of dissatisfaction differ by race/ethnicity, that would be a provocative finding, even if it is only preliminary.

Reviewer #2: Overall this is a good paper on an important topic. I do have a few thoughts.

1. On page 9 lines 173-174. The authors describe how they excluded participants were missing data on the perceived discrimination measure. However, page 13 second paragraph you wrote that you calculated using values and the perceived discrimination scale by using the mean value of the nine missing items. Which one did you do?

2. On page 18, there is no discussion of pharmacy is the difference between no PD and any PD there was significant at the .001 level

3. on page 22 line 423, the word explain should be changed to identify

Which

6. PLOS authors have the option to publish the peer review history of their article (what does this mean?). If published, this will include your full peer review and any attached files.

Reviewer #1: No

Reviewer #2: No

---

## [Author Response · Author response to Decision Letter 0]

13 Jul 2020

Editor Comments

Please pay particular attention to Reviewer 1's comments about clear explanation of the methodology, and clarity and appropriateness of inference.

Response: We have made substantial revisions to the paper in response to Reviewer 1’s comments. Specific changes made in response to each comment are itemized below and changes to the manuscript reflecting these suggestions appear in bolded text.

Reviewers’ Comments

Reviewer 1, Comments to the Author

1. This manuscript presents findings from a national, cross-sectional survey of VA patients stratified by race/ethnicity (non-Hispanic white vs Hispanic vs black). Authors code open-ended telephone survey responses and then conduct regression analyses to identify correlates of perceived discrimination overall and by veteran race/ethnicity. The focus on veterans with pain is a strength, because racial/ethnic disparities in pain management have been well documented and pain is a common topic of patient dissatisfaction. As detailed below, there are several areas that dampen my enthusiasm: the study design is not clear early on to readers, a clearer focus on implications for pain management (including possibly analyzing pain as a covariate) and more tempered conclusions appropriate to the limitations of cross-sectional data. If these concerns can be addressed, the manuscript has potential to make a meaningful contribution to the literature and advance knowledge.

Response: We appreciate the reviewer’s overall assessment of the potential contribution of our paper. As detailed below, we have made changes throughout the paper to clarify the study design (see comments 2, 3, and 6), provide the implications of our findings for pain management (see comments 5, 7, and 9), and temper the conclusions we draw given the limitations of our cross-sectional data (see comments 4, 19, 20, and 21).

Reviewer 1, Specific Comments

2. Title: Overall study design and procedures are not clear based on the abstract and intro; I re-read abstract several times and it was not clear until well into methods section that this study is a survey with both closed-ended items and (semi-structured) open-ended interview questions. Emphasizing the “survey” and de-emphasizing the “mixed methods” aspect throughout manuscript will make study design clearer for readers. Specifically, specifying the population and study design in title will decrease potential for reader confusion. Title should be, “Identifying healthcare experiences … among veterans with pain: a cross-sectional mixed methods survey.”.

Response: We have adopted the suggested title and changed “interview” to “survey” in the abstract and throughout the paper in response to this comment. 

3. Abstract: Edit to make study design clearer for readers. In DESIGN, consider replacing the term “telephone interviews” with “telephone survey.” In MAIN MEASURES, make it clearer that perceived discrimination (dependent variable) and open-ended questions (which were coded and transformed to derive independent variables) were both collected during the same survey interview. I appreciate the mixed methods aspect but the quantitative component is predominant in this study and should drive how study is described.

Response: As suggested, we have removed the term mixed methods from the abstract and now refer to the interviews as surveys. We carried these changes throughout the rest of the manuscript (e.g., methods, tables, discussion) to make the design clearer to readers. We also revised the methods in the abstract and main text to clarify that the perceived discrimination survey and the open-ended questions were collected in the same survey.

4. Abstract: Final sentence of CONCLUSION needs to be more qualified / cautious. It is too strong to conclude definitively from this study that findings “underscore the need for interventions…” In addition, implications may be specific to veterans with pain.

Response: We appreciate the reviewer’s caution that our study does not point to specific interventions because causal relationships cannot be established using cross-sectional data. We also think that shying away from any implications is overly cautious and could undermine the potential impact of our findings. We therefore have toned down the definitive nature of our concluding statement while still calling for research that explore possible interventions to reduce perceived discrimination among patients. The final sentence now reads:

Future studies should test whether interventions targeting these domains reduce patient perceptions of racial/ethnic discrimination in healthcare.

5. Introduction, next-to-last paragraph, page 6: Intro discusses the need to understand correlates of perceived discrimination to improve care generally. However, given the known racial/ethnic disparities in pain management, study findings specifically have potential to generate hypotheses/insights that can help to address racial/ethnic disparities in pain management. Making this point explicitly in the intro (and briefly citing literature on racial/ethnic disparities in pain mgmt) would strengthen study justification and make study more compelling for readers.

Response: Thank you. We added the following sentence to the introduction (lines 126-129).

Given the known racial/ethnic disparities in pain management [14–17], we then looked at associations between the dissatisfaction domains and the quantitative measure of discrimination within White, African American, and Latino groups to gain insights into potential targets for disparity interventions.

We also added the following sentence to the first paragraph of the discussion to put the study in context of the broader literature on racial disparities in pain management and the likely role of discrimination as a factor contributing to those disparities (lines 482-484):

In addition, there are well-documented racial/ethnic disparities in both the clinical impact and treatment of pain, for which discrimination may be a contributing factor [14,17,32].

6. Introduction, final paragraph: Again, study design is difficult to discern from this paragraph. Describing the survey design before the qualitative coding and using the term “mixed methods survey” instead of “mixed methods study” will help clarify design for readers.

Response: We removed the term mixed methods from the first sentence of the paragraph and replaced mixed methods study with mixed methods survey later in the paragraph.

7. Participants, page 9: Adding more detail about patient pain would be helpful given the study focus on patients w pain. It is also plausible that pain & pain management would differ for veterans with vs without perceived discrimination. Did parent study collect data on pain (eg pain numeric rating scale, PEG, pain type/location; study mentions that DISC collected administrative VHA data)? If so, pain-specific variables should be included as covariates or at least summarized in Table 2. If parent study did not collect data on patient pain, this should be acknowledged as a limitation. Did the QOL measures include pain-specific items that could be reported separately?

Response: The parent study was focused on primary care patients and therefore did not collect pain measures. We note this as a limitation in the discussion (lines 593-595):

A second limitation is that, because the larger study was about patient satisfaction in general and targeted primary care patients, the survey did not include clinical measures of pain.

8. Page 10: Specify whether the coders working in teams of two independently applied the master codebook to each interview or whether they coded together as a team.

Response: We have clarified that coders independently coded the interviews and then met to adjudicate any discrepancies that emerged between them (lines 190-194):

Working initially in teams of two, coders applied the final master codebook to their assigned recordings, each completing the coding independently. The two coders then engaged in an inter-coder reliability adjudication process where they deliberated in order to come to agreement per code. Twenty percent of the interviews were coded using an inter-coder reliability process.

9. Table 1: This is a nice table. Since “unresolved pain” is a specific code within the quality of care domain, it would be helpful for results to specify proportion of patients for whom this code was present. This might be reported in table 3 or in the manuscript where table 3 is discussed. would also be helpful to note in parentheses after each example the code(s) that example represents within each domain.

Response: Thank you for this excellent suggestion. In response, we revised our analyses to treat the unresolved pain code as its own predictor variable rather than grouping it with the other codes in the quality of care domain. This change resulted in changes to all of the results, tables, and Figure 2. This change also impacted the discussion section, which now comments on the new finding that the association of unresolved pain with perceived discrimination was only observed among White participants (lines 510-518). We have also added the codes for each quote included in Table 1. 

10. Table 1: All the examples under “staff demeanor” relate to pain. Since demeanor was one of the significant correlates of perceived discrimination, it would be helpful to describe in results section the prevalence of pain-specific codes for this domain (either quantitatively, qualitatively, or both). A parallel analysis for the other significant domain (interaction w staff) would also be helpful. This additional detail would help convey to readers how salient pain-related comments are among the survey responses, and by extension, the extent to which pain-specific concerns may correlate w perceived discrimination.

Response: Unfortunately, the sub-analysis suggested by the reviewer is not possible because there are no pain-specific codes in the staff demeanor or interactions domains. The only code that obviously referenced pain is “unresolved pain”. We addressed this comment in three ways:

1) We added new example quotes to Table 1 to show the diversity of interview responses. 

2) To address the reviewer’s request, one author (SZ) reviewed a 20% random sample of quotes for the domains of dissatisfaction with staff demeanor and negative interactions to see if pain-specific concerns emerged and to determine if the requested sub-analysis would be fruitful. In these domains, only 3.8% of staff demeanor quotes (4 of 104 reviewed) and only 5.2% of negative interactions quotes (8 of 155 reviewed) clearly referenced pain concerns. A full analysis of pain-specific quotes within these domains was not pursued further due to low frequency.

3) As noted in our response to Comment 9, we reran our statistical models with unresolved pain as a predictor that was separate from the other quality of care codes (see Tables 3-5). Indeed, poor care attributed to unresolved pain was associated with perceived discrimination among White patients. We describe these new findings throughout the results and discussion.

11. Pages 20-21: Very interesting that sub-analyses found that “interaction w staff” was correlated w dissatisfaction only for Hispanic vets, while “demeanor” was correlated w dissatisfaction only for black vets. However, on page 20-21 all the examples or “interaction w staff” came from black vets while almost all the examples of “demeanor” came from Hispanic vets. Why is this?

Response: We thank the reviewer for bringing this oversight to our attention. We did not systematically select quotes based on patient race or ethnicity and had not realized the pattern noted by the reviewer for our selected quotes. In response to this comment, we re-examined our quotes and selected alternative examples from all three racial/ethnic groups in our sample (lines 395-441).

12. Related to point 11 above: More detailed results about the differences in findings by veteran race/ethnicity would strengthen the study. As mentioned in the intro, discrimination experienced by Hispanic vets is understudied relative to black vets, so including large proportion of Hispanic vets is a particular strength of this study. Although Tables S1 and S3 present some differences by veteran race/ethnicity and there are obvious differences by race/ethnicity, these differences are mostly glossed over in the main manuscript. For Table S3, may be helpful to see differences across all the themes (not just top 3). More nuanced discussion of qualitative and quantitative differences by race/ethnicity (including differences that may not have met the very strict corrected p value threshold) would help generate hypotheses about how perceived discrimination differs for black vs Hispanic vs NH white veterans, which in turn would increase study impact.

Response: We appreciate the reviewer’s recognition that the inclusion of Latino Veterans in our sample as a strength of the study. Given the exploratory nature of our quantitative analysis, we are also careful not to over-interpret differences across groups that could be driven by relatively low frequency of certain codes within certain groups. Our use of the Bonferroni correction helps focus the discussion on stronger associations that are less likely to be due to chance. That said, in response to the above comment, we have moved Tables S1 and S3 to the main text as Tables 5 and 6, respectively, so that readers have easier access to the results stratified by race/ethnicity. We also report the race-specific results in the abstract and have expanded Table S2 (now Table 6) to include all the codes within the domains of interactions with staff and staff demeanor. Finally, we have added text in the results and discussion to highlight some of the nuances across the groups. 

 Results (lines 442-452):

 The patterns of codes were generally similar across racial/ethnic groups with some subtle nuances (Table 6). For example, many of the codes were more frequent among White participants than among African American or Latino participants, especially rudeness, not listening, treating Veterans like a number, general statements about unresolved pain, and being perceived as drug-seeking. Also, the distribution of individual codes pertaining to staff demeanor were different among African American and Latino participants. For example, stigma, dishonest/untrustworthy, and unhelpful codes came up more frequently for African American participants, whereas the relatively infrequent codes of unprofessional and uninviting/unwelcoming demeanor came up more often for Latino participants (Table 6).

 Discussion (lines 491-518):

We observed variation in the types of domains that were associated with perceived discrimination across racial/ethnic groups, with dissatisfaction in interactions with staff being associated with perceived discrimination among Latinos and dissatisfaction with staff demeanor being associated with perceived discrimination among African Americans. The overarching pattern of negative experiences in interpersonal domains for African American and Latino racial/ethnic groups support conclusions from prior qualitative studies that perceptions of discrimination in healthcare among patients from minority populations manifest, in large part, from poor interpersonal interactions with clinical and non-clinical staff [10–12]. Our results are also consistent with research on microaggressions, which are commonplace indignities or insults that are directed, often unintentionally, at members of marginalized groups [33,34]. The types of interpersonal interactions associated with perceived discrimination among African American and Latino patients in the current study suggest that patients were experiencing microaggressions from healthcare employees [35]. It is noteworthy that, even though negative interactions with staff and encountering staff with negative demeanor were equally or more prevalent among White patients, such interpersonal slights were only associated with perceived discrimination among minority patients.

 Another nuance observed in this study was that unresolved pain was associated with perceived discrimination only among White patients. One interpretation of this novel finding is that stigma and frustration surrounding pain management may be a salient issue for White patients, whereas negative interpersonal interactions with healthcare employees in general may be more salient experiences among African American and Latino patients. Given the exploratory nature of this mixed methods analysis, additional research is needed to better understand the nuances in themes we observed as correlates of perceived discrimination across racial/ethnic groups. 

13. Finally, related to point 12 above: manuscript needs to report more detail about perceived discrimination by NH white veterans and the factors/domains correlated with perceived discrimination in these patients compared to Hispanic and black veterans. Introduction (page 6) makes a point to cite prior studies about experience of “reverse discrimination” and states that discrimination can be perceived by vets of any race/ethnicity. However, results and discussion sections only focus on Hispanic and black veterans.

Response: In response to comment 11 (above), we examined unresolved pain and found this was a significant correlate of perceived discrimination among White Veterans. We now discuss this finding in the abstracts, results, and discussion. We also include sample quotes from all three groups, including White Veterans, in the results section. 

14. Discussion, page 22: Can authors put the prevalence of any perceived discrimination in their sample (37.5%) in context? Is this higher, lower, or similar to rates reported in other studies of veterans or non-veterans? A quick pubmed search found some articles that may be relevant (e.g. Sorkin, D. H., et al. "Racial/ethnic discrimination in health care: Impact on perceived quality of care." Journal of General Internal Medicine 25(5): 390-396).

Response: We have added the following text and references to note that the prevalence observed in our sample is within the wide range of prevalence observed in other studies of Veterans. Due to the wide variation in prevalence and inconsistencies in the specific measures used to assess perceived discrimination across different studies, we do not think it would be appropriate to provide a specific range within the text. We instead point readers to papers that they can read if they are interested in learning more about variation in perceived discrimination across different populations (lines 552-559):

The prevalence of perceived discrimination observed in this study is within the wide range observed in prior studies that have assessed perceived racial/ethnic discrimination in VA healthcare settings [4,5,37–39]. Although the overall prevalence varies across studies depending on the targeted patient population and the measure used to assess perceived discrimination [5], a common pattern observed across the literature is that racial/ethnic discrimination is more frequently reported by racial/ethnic minority patients compared to White patients, as was the case for our study.

15. Second sentence of discussion, line 423: “discrimination” should be “perceived discrimination.”

Response: Thank you for catching this omission. We have changed the terminology as suggested (now line 485).

16. Third sentence of discussion, line 428: Both references (ref 8, 24) are specific to patients with sickle cell disease. References more relevant to this study population would be helpful.

Response: We have supplemented the original references with papers showing similar relationships between experiences of discrimination and pain intensity, disability, and development of chronic pain (lines 478-484):

It is important to understand experiences of perceived discrimination in healthcare settings for patients with pain, specifically, as experiences of discrimination have been linked to increased pain sensitivity, pain severity, disability, and development of chronic pain [8,9,28–31].

17. Discussion, page 22-23: The finding that interpersonal interactions are consistent drivers of perceived discrimination across race/ethnicity is very interesting and important. I appreciate that the “overarching pattern of negative experiences in interpersonal domains … [was similar for] all racial/ethnic groups.” However, more detail about the specific differences and variation across the 3 racial/ethnic groups studied should be included in discussion, even if similarities outweigh differences. As noted above, the intro highlighted the inclusion of Hispanic vets and potential ‘reverse discrimination’ among NH white vets as strengths of this study. Thus, readers will be expecting results and discussions related to these two topics.

Response: Thank you. We now discuss the specific findings correlating with reverse discrimination among non-Hispanic white Veterans. For more on this, see our detailed response to comment 13 above. 

18. Discussion, page 23 line 452: Please replace the term “soft skills” with communication skills, remove the scare quotes, and remove communication skills from the list at the start of that sentence. All the examples provided in this paragraph are specific types of communication skills. It is inappropriate to use terminology that implies communication skills are less important or more subjective than other clinical skills.

Response: We have modified the sentence as suggested by the reviewer (lines 544-549):

Such strategies could include expanded provider training in pain management and addiction stigma, as well as active listening and other communication skills among the healthcare delivery workforce. Other strategies could involve institutional changes to provide more time in the clinical encounter for pain management, and to support respectful and empowered communication between healthcare employees and patients.

19. Discussion: On page 25, the authors appropriately discuss limitations of this cross-sectional survey, particularly inability to assess causation or control for unmeasured confounding. However, the paragraph starting on line 443 (pages 23-24) seems to ignore these limitations by suggesting that study results directly support specific future interventions which all implicitly assume that experiencing the themes identified causes perceived discrimination. This paragraph should be much more cautious about results interpretation and next steps and include discussion of both potential ‘causal’ and ‘non-causal’ explanations/interpretations of study findings. Based on data presented, reverse causation is also plausible (i.e. patients who have experienced past discrimination are more likely to report/remember/perceive discrimination in the VA).

Response: We have substantially edited this paragraph with more tentative language (lines 532-549):

While the study design does not permit causal inference, our thematic analysis of the domains that were associated with perceived discrimination in this study (i.e., staff interactions, staff demeanor, unresolved pain), highlight areas deserving of future research and potential intervention. First, African American patients who perceived discrimination were likely to have interactions with healthcare employees who were rude, did not listen to patients, or did not provide the information patients desired. Latino patients who perceived discrimination were likely to describe interpersonal interactions with healthcare employees who appeared unconcerned about the patient’s wellbeing, made the patient feel stigmatized, or were untrustworthy. Additionally, White patients who perceived discrimination felt they were being accused of illicit behavior and substance abuse. Future research is needed to determine if targeting these aspects of interpersonal interactions improve patient experiences or reduce perceptions of discrimination in healthcare settings. Such strategies could include expanded provider training in pain management and addiction stigma, as well as active listening and other communication skills among the healthcare delivery workforce. Other strategies could involve institutional changes to provide more time in the clinical encounter for pain management, and to support respectful and empowered communication between healthcare employees and patients. 

20. Page 23, the sentence starting on line 454 (“Disruptive innovations in healthcare models… should also be explored.”) should be deleted. Even setting aside the inability of this study to evaluate causal associations, the data presented does not support or suggest that these types of interventions are likely to reduce perceived discrimination. If anything, the data suggest such “disruptive” system-level innovations would not reduce perceived discrimination, because systems domains (facilities, nonmedical aspects of care, continuity of care, access) were not associated w perceived discrimination in the final model.

Similarly, the two sentences starting on page 465 (“not only should such initiatives…in this study”) should be deleted because they do not follow from the data presented and implicitly interpret correlations as causal associations. The study did not address unconscious bias or beliefs about biological difference between races, or how presence of these staff attitudes might relate to the qualitative themes.

Response: We substantially revised the discussion to include more tentative language while still suggesting topics to pursue in subsequent research. The paragraph referenced in the above comment now reads as follows (lines 561-575):

It is not possible from the current study to determine if stigma related to addiction, race/ethnicity, or other factors are direct causes of perceived discrimination. And yet, the high prevalence of stigma themes among persons receiving treatment for pain underscores the magnitude of this problem in healthcare. Additional work is needed to identify effective strategies to reduce stigma for patients with pain. Some promising avenues for future research might include 1) initiatives that promote principles of diversity and inclusion across all aspects of healthcare delivery [41], 2) education for providers and staff in unconscious biases [42] and misperceptions about biological differences between racial/ethnic groups among healthcare providers [32], and 3) deconstructing institutionalized processes that contribute to differential power, privilege, and outcomes across groups [43]. It is also important for future research in this area to determine if interventions aimed at healthcare discrimination could ameliorate persisting racial/ethnic disparities in pain management and outcomes. 

21. As noted in prior comments, it would strengthen discussion to talk about whether findings and further efforts to reduce perceived discrimination among veterans has the potential to ameliorate documented racial/ethnic disparities in pain management.

Response: We added the following sentence to highlight this research gap (lines 572-575):

It is also important for future research in this area to determine if interventions aimed at healthcare discrimination could ameliorate persisting racial/ethnic disparities in pain management and outcomes.

22. Authors have already published data from parent study on positive patient comments (JGIM 2018;33:305-331), where they found no consistent differences in correlates of satisfaction by veteran race/ethnicity. Would be helpful to briefly mention the results of that analysis to put the current findings related to negative comments in context. If correlates of veteran satisfaction are similar across racial/ethnic groups but correlates of dissatisfaction differ by race/ethnicity, that would be a provocative finding, even if it is only preliminary.

Response: We now describe our findings in relation to prior published data from the parent study (lines 519-531):

While an earlier paper published from the DISC cohort has reported on quantitative ratings of healthcare satisfaction [13], the current mixed methods analysis draws on rich qualitative codes to help explain drivers of perceived discrimination in healthcare for patients receiving pain management. At first glance, the higher prevalence of perceived discrimination among African American Veterans is at odds with previously published DISC findings of few racial/ethnic differences in quantitative ratings of healthcare satisfaction. As in prior work, we find that Likert satisfaction ratings alone do not fully capture adverse healthcare experiences such as those perceived as discriminatory [12,36]. The qualitative data analyzed in the current study provide a richer lens to understand the more nuanced aspects of healthcare encounters that contribute to negative experiences, including those attributed to one’s race/ethnicity.

Reviewer 2, Specific Comments

23. Page 9, lines 173-174: The authors describe how they excluded participants were missing data on the perceived discrimination measure. However, page 13 second paragraph you wrote that you calculated using values and the perceived discrimination scale by using the mean value of the non-missing items. Which one did you do?

Response: We thank the reviewer for pointing out the need for clarification. We note in the participants section that we excluded from analysis participants with substantial missing data on the primary outcome measure (i.e., 2 or more items). We also revised the description of our primary outcome to remind readers of the exclusion criteria and to indicate that, for small missingness, we calculated discrimination as the average of non-missing items.

Participants (lines 166-171):

 The current study focused on White, African American, and Latino DISC participants who met the following additional criteria: 1) responded “yes” to the question, “Have you received pain management from the VA in the last 24 months?”; 2) reported on their satisfaction with pain management; 3) and completed a measure of perceived discrimination (described below). We excluded participants missing data for >2 items on the 7-item perceived discrimination measure.

Primary outcome (lines 243-246): 

For participants answering at least 6 of the 7 items, we calculated an overall discrimination score as the average of non-missing items; participants missing 2 or more items were excluded.

24. Page 18: there is no discussion of pharmacy in the difference between no PD and any PD there was significant at the .001 level.

Response: We note in the text that dissatisfaction with pharmacy services occurred more frequently among patients with perceived discrimination compared to those with no perceived discrimination when discussing Table 3. We do not elaborate on this variable further because it was no longer statistically significant after controlling for the other healthcare domains (Tables 4 and 5). 

25. Page 22, line 423: the word explain should be changed to identify.

Response: We have made the requested change (now line 476).

---

## [Decision Letter · Decision Letter 1]

31 Jul 2020

Identifying healthcare experiences associated with perceptions of racial/ethnic discrimination among veterans with pain: A cross-sectional mixed methods survey

PONE-D-20-03894R1

Dear Dr. Hausmann,

We’re pleased to inform you that your manuscript has been judged scientifically suitable for publication and will be formally accepted for publication once it meets all outstanding technical requirements.

Kind regards,

M Barton Laws

Academic Editor

PLOS ONE

Additional Editor Comments (optional):

Although I am recommending acceptance and do not feel this needs further peer review, please note that both reviewers have made minor queries. You will probably want to address them in the final version of your paper. That said, I am very pleased to see this work be published.

Reviewers' comments:

Reviewer's Responses to Questions

**Comments to the Author**

1. If the authors have adequately addressed your comments raised in a previous round of review and you feel that this manuscript is now acceptable for publication, you may indicate that here to bypass the “Comments to the Author” section, enter your conflict of interest statement in the “Confidential to Editor” section, and submit your "Accept" recommendation.

Reviewer #1: All comments have been addressed

Reviewer #2: All comments have been addressed

2. Is the manuscript technically sound, and do the data support the conclusions?

Reviewer #1: Yes

Reviewer #2: Yes

3. Has the statistical analysis been performed appropriately and rigorously? 

Reviewer #1: Yes

Reviewer #2: Yes

4. Have the authors made all data underlying the findings in their manuscript fully available?

Reviewer #1: No

Reviewer #2: Yes

5. Is the manuscript presented in an intelligible fashion and written in standard English?

Reviewer #1: Yes

Reviewer #2: Yes

6. Review Comments to the Author

Reviewer #1: This revised manuscript is substantially improved compared to the initial submission. I appreciate the amount of work the authors put in to this revision. In addition to addressing all my comments, they re-ran analyses using unresolved pain as a separate code. The revised results and revised discussion are now much more interesting. The discussion about differences in experiences associated with perceived discrimination among white, Latino, and black veterans is good, and in particular the finding that unresolved pain was associated with perceived discrimination (particularly among white veterans) was very interesting. The added tables are helpful. This paper represents a useful addition to the literature and should be of interest to PLOS ONE readers.

One minor comment - line 124 mentions 9 qualitative domains; the rest of the manuscript mentions 10 domains. please check whether mention of 9 domains on line 124 is a typo.

Reviewer #2: I could not find any description of the personnel who conducted the telephone surveys. Are they matched to the ethnicity of the participants? There should be some discussion about this and its potential to bias responses from participants.

7. PLOS authors have the option to publish the peer review history of their article (what does this mean?). If published, this will include your full peer review and any attached files.

Reviewer #1: No

Reviewer #2: No

---

## [Editor Report · Acceptance letter]

25 Aug 2020

PONE-D-20-03894R1 

Identifying healthcare experiences associated with perceptions of racial/ethnic discrimination among veterans with pain: A cross-sectional mixed methods survey 

Dear Dr. Hausmann:

I'm pleased to inform you that your manuscript has been deemed suitable for publication in PLOS ONE. Congratulations! Your manuscript is now with our production department. 

Kind regards, 

on behalf of

Dr. M Barton Laws 

Academic Editor

PLOS ONE